# Imaging moiré deformation and dynamics in twisted bilayer graphene

Tobias A. de Jong [1✉], Tjerk Benschop [1], Xingchen Chen[1], Eugene E. Krasovskii [2,3,4], Michiel J. A. de Dood [1], Rudolf M. Tromp[1,5], Milan P. Allan[1] & Sense Jan van der Molen [1✉]

In 'magic angle' twisted bilayer graphene (TBG) a flat band forms, yielding correlated insulator behavior and superconductivity. In general, the moiré structure in TBG varies spatially, influencing the overall conductance properties of devices. Hence, to understand the wide variety of phase diagrams observed, a detailed understanding of local variations is needed. Here, we study spatial and temporal variations of the moiré pattern in TBG using aberration-corrected Low Energy Electron Microscopy (AC-LEEM). We find a smaller spatial variation than reported previously. Furthermore, we observe thermal fluctuations corresponding to collective atomic displacements over 70 pm on a timescale of seconds. Remarkably, no untwisting is found up to 600 °C. We conclude that thermal annealing can be used to decrease local disorder. Finally, we observe edge dislocations in the underlying atomic lattice, the moiré structure acting as a magnifying glass. These topological defects are anticipated to exhibit unique local electronic properties.

[1] Leiden Institute of Physics, Leiden University, P.O. Box 9504, 2300 RA Leiden, The Netherlands. [2] Departamento de Polímeros y Materiales Avanzados: Física, Química y Tecnología, Universidad del Pais Vasco UPV/EHU, 20080 San Sebastián/Donostia, Spain. [3] IKERBASQUE, Basque Foundation for Science, E-48013 Bilbao, Spain. [4] Donostia International Physics Center (DIPC), E-20018 San Sebastián, Spain. [5] IBM T.J.Watson Research Center, 1101 Kitchawan Road, P.O. Box 218, Yorktown Heights, New York, NY 10598, USA. ✉email: jongt@physics.leidenuniv.nl; molen@physics.leidenuniv.nl

In twisted bilayer graphene (TBG) a moiré pattern forms that introduces a new length scale to the material. At the "magic" twist angle $\theta_m \approx 1.1°$, this causes a flat band to form, yielding emergent properties such as correlated insulator behavior and superconductivity[1–4]. In general, the moiré structure in TBG varies spatially, influencing the local electronic properties[5–9]. This has clear consequences for charge transport experiments, where a percolative average of the microscopic properties is measured. Local variation in twist angle and strain in TBG will directly influence the result of such experiments. In particular, to understand the wide variation observed in the phase diagrams and critical temperatures, a more detailed understanding of the local moiré variation is needed[10].

However, imaging such microscopic variations is non-trivial. A myriad of experimental techniques has been applied to the problem[11–17], each only resolving part of the puzzle due to practical limitations (capping layer or device substrate, surface quality, or measurement speed).

Here, we use aberration-corrected low energy electron microscopy (AC-LEEM)[18,19], which measures an image of the reflection of a micron-sized beam of electrons at a landing energy $E_0$ (0–100 eV, referenced to the vacuum energy) in real space, in reciprocal space (diffraction), or combinations thereof. This allows us to perform large-scale, fast, and non-destructive imaging of TBG, including device-scale moiré images and dynamics on timescales of seconds. In addition, spectroscopic measurements, yielding information on the material's unoccupied bands can be done by varying $E_0$[20,21].

Using AC-LEEM to image moiré patterns enables high-temperature imaging and has the benefit no suspended samples are required like they are for TEM-based techniques. This means that sample geometries closely resembling device geometries can be imaged, including devices with leads. Even though not shown here, imaging the moiré pattern through a capping layer of hBN would be possible, although this would be limited to a very thin capping layer of at most a few atomic layers (where SEM-based techniques have demonstrated imaging through much thicker layers[16]).

At 500 °C, we observe thermal fluctuations of the moiré lattice, corresponding to collective atomic displacements of less than 70 pm on a time scale of seconds[22]. Despite previous concerns, no untwisting of the layers is found, even at temperatures as high as 600 °C[23,24]. Finally, we report the existence of individual edge dislocations in the atomic and moiré lattice.

## Results and discussion
A schematic of a TBG sample as used in this work is shown in Fig. 1a. In Fig. 1b, LEEM spectra are shown, taken on several locations of such a TBG sample. These LEEM spectra are directly related to layer count, as described in refs.[21,25–27]; on the one hand via interlayer resonances in the 0–5 eV range, on the other hand via the gradual disappearance of a minimum at 8 eV. Here, more graphene layers (having a band gap at 8 eV) are progressively masking an hBN band underneath. This allows us to determine the local graphene layer count for each point on our sample. To visualize that, we choose three characteristic energies, i.e., $E_0 = 4$ eV (red), $E_0 = 8$ eV (green), and $E_0 = 17$ eV (blue) (see Fig. 1b), and combine stitched overviews at these energies into a single false-color image (Fig. 1c). This overview confirms that the sample consists of large TBG areas (darker green in Fig. 1c) surrounded by monolayer graphene (pink), on an hBN flake (blue/purple) on silicon (black). Stripes of brighter green indicate areas of 2-on-2 graphene layers (lower stripe), 2-on-1 (upper stripe), and 1-on-2 (wedge on the lower right). The relatively homogeneous areas are themselves separated by folds, appearing

as black lines. The folds locally combine in larger dark nodes (confirmed by AFM, see Supplementary Fig. 11). A few folds, however, have folded over and appear as lines of higher layer count. Hence, Fig. 1c provides a remarkable overview of a larger-scale sample, with detailed local information.

Increasing $E_0$ beyond 25 eV, stacking boundaries and AA-sites become visible[25,28]. This is consistent with ab initio calculations of LEEM spectra for different relative stackings, as presented in Fig. 1d. Therefore, imaging at $E_0 = 37$ eV (indicated in Fig. 1d) yields a precise map of the moiré lattice over the full TBG area (see Fig. 1e). We find that separate areas, between folds, exhibit different moiré periodicities and distortion[29]. This allows us to study different moiré structures on a single sample. Fig. 1f–i shows full resolution crops of areas indicated in Fig. 1e. The observed twist angles on this sample range from <0.1° to 0.7°. For smaller angles, we observe local reconstruction towards Bernal stacking within the moiré lattice, consistent with literature[12,15]. The best resolution was reached on another sample with a twist angle of 1.3° (See Supplementary Fig. 3).

**Distortions and strain**. The moiré patterns show distortions, corresponding to local variations in twist angle and (interlayer) strain. Near folds, for instance, the strain increases resulting in strongly elongated triangles, for example in the lower right corner of Fig. 1e[30]. Despite their relative homogeneity, the moiré areas in Fig. 1f–i also show subtle distortions. As structural variations correlate directly with local electronic properties, we will quantify them in detail[5,6]. For this, we use adaptive geometric phase analysis (GPA), extending our earlier work on STM data of moiré patterns in TBG (see Supplementary Note 2)[31–35]. This method, illustrated in Fig. 2a–c, yields the displacement field with respect to a perfect lattice, by multiplying the original image with complex reference waves followed by low-pass filtering to obtain the GPA phase differences, which are then converted to the displacement field. This field fully describes the distortion of the moiré lattice and allows us to extract key parameters such as the local twist angle $\theta^*(r)$ (see Fig. 2d), and heterostrain magnitude $\epsilon(r)$ and direction (see Fig. 2e)[33,36] The distortions of the moiré pattern correspond directly to distortions of the atomic lattices, magnified by a factor $1/\theta$ and rotated by $90° + \theta/2$[33,37].

The extracted variation in twist angle and heterostrain for various regions of the sample, including those in Fig. 1f–i, is summarized in Fig. 2f, g, respectively. The twist angle variation within each domain is much smaller than the variation in twist angle between the separate, fold-bounded areas. Within domains, standard deviations range from 0.005° to 0.015°, i.e., significantly smaller (by a factor 3–10) than previously reported[13,15,33]. The strain observed is around a few tenths of a percent, which is considerable. In some domains, we find an average strain of the atomic lattice of up to $\epsilon = 0.4\%$. According to earlier theoretical work, such values are high enough to locally induce a quantum phase transition[8].

The variation of $\epsilon$, as for $\theta^*$, within the domains is significantly lower than in earlier studies. We do note that the use of GPA introduces a point spread function (PSF) that is broader than the PSF of the instrument, resulting in a lower displacement field frequency response at small scales and therefore a somewhat reduced variation. Nevertheless, the combined PSF of instrument and analysis is still comparable to other techniques that do not image the unit cell directly, allowing for a direct comparison.

We hypothesize that the difference in variations with literature stems from the relatively high temperature to which we heated the sample and measured it, combined with the relatively long averaging time of this measurement (≥16 s for all data in Fig. 2). The high temperature induces thermal fluctuations of the lattice

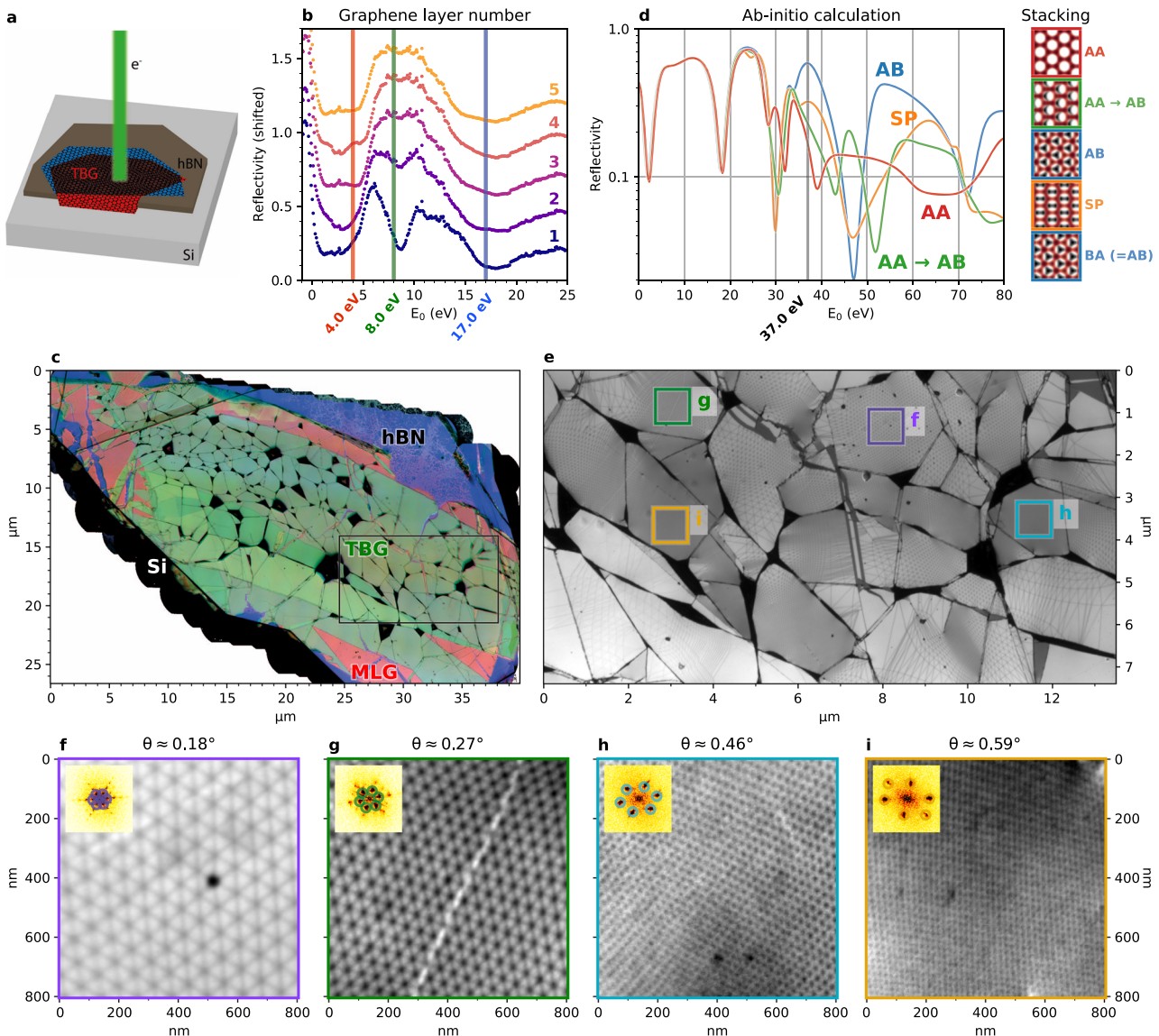

**Fig. 1 Device-scale imaging of TBG. a** Schematic of the sample, with two twisted graphene flakes (TBG) on top of a hexagonal boron nitride (hBN) flake on a silicon (Si) substrate. **b** Local spectra that were used to determine the graphene layer count. Vertical lines indicate the imaging energies used for panel (**c**), the number of graphene layers for each spectrum is indicated on the right. **c** Stitched composite bright field overview of a sample using 4, 8, and 17 eV as imaging energies in red, green, and blue respectively (see main text for color interpretation). Visible defects include folds in black, tears, where the monolayer (red/pink, indicated as MLG) or even bare hBN (blue, purple) shines through, bubbles (bright), and some polymer residue in the lower and upper right (dark speckles). The black rectangle indicates the area shown in panel (**e**). **d** Ab initio calculations of LEEM spectra of different relative stackings of bilayer graphene, 37 eV indicated. **e** Stitched bright field overview of the same sample imaged at $E_O = 37$ eV, for optimal stacking contrast, revealing the moiré patterns. **f–i** Crops of different twist angle areas from (**e**). Insets show Fourier transforms and the detected moiré peaks, with the average twist angle $\theta$ extracted from those indicated. All data shown in the main text have been collected from the sample represented in panel (**c**).

(as demonstrated below), allowing the system to approach a more homogeneous, lower energy, state.

**Edge dislocations**. So far, we have discussed structural properties varying on the moiré length scale. However, the moiré magnification of deformations is general and extends to atomic edge dislocations (visualized in Fig. 3a). This type of topological defect stems from a missing row of atomic unit cells and is characterized by an in-plane Burgers vector (in red)[38,39]. The addition of a second (twisted) atomic layer magnifies (and rotates) the defect to an edge dislocation in the moiré lattice (illustrated in Fig. 3d, e)[37]. In all cases, the location of the defect can be pinpointed by a singularity in the GPA phases and characterized by a Burgers vector (Fig. 3b, e)[40].

The movement of edge dislocations in single-layer graphene and their interaction with both the in-plane and out-of-plane deformations of the atomic lattice have been studied extensively using TEM[40–42].

In our sample, we find a few such defects in the moiré lattice (see Supplementary Figs. 8 and 9). In Fig. 3f, g, we show an edge dislocation in a topographically flat region with $\theta = 0.63°$ (AFM data in Supplementary Note 1). The absence of any visible out-of-plane buckling in AFM suggests the dislocation, which in the freestanding case would be buckled[41], is flattened out by vdW adhesion between the layers and to the hBN substrate. Contrary to TEM, the low-energy electrons used for imaging here do not sputter carbon atoms, preventing the creation of edge dislocations pairs and impairing movement. We do not observe the creation

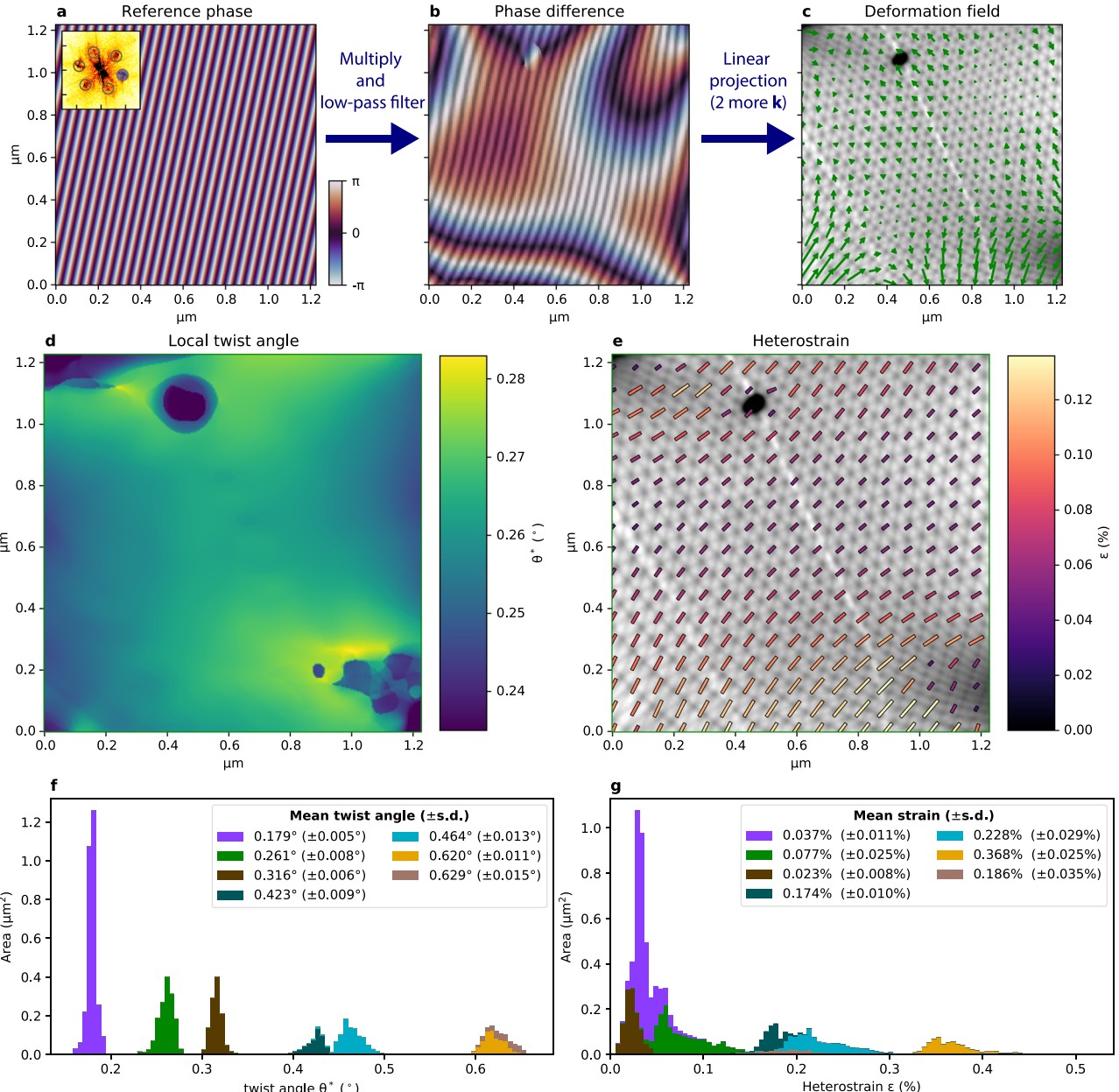

**Fig. 2 Distortion variation from geometric phase analysis. a** Reference phase corresponding to one *k*-vector used in GPA, as extracted from the Fourier transform of the corresponding image (inset, extracted *k*-vectors are indicated by black circles). **b** Phase difference or GPA phase, obtained by multiplying with the original image and low-pass filtering. Overlayed is the corresponding extracted image wave in gray scale. **c** The displacement field extracted from the GPA phase represented in panel b and those of 2 more *k*-vectors. **d** Extracted local twist angle $\theta^*$. **e** Extracted local heterostrain. The length and color of bars indicate the magnitude $\epsilon$ of the heterostrain, direction is the direction of elongation of the atomic lattice. **f, g** Distribution of $\theta^*$ and $\epsilon$ extracted from different areas on the sample. Bar colors correspond to colors in Fig. 1e, with the remaining areas shown in Supplementary Fig. 1.

or annihilation of edge dislocation pairs in the microscope, even at elevated measurement temperatures (500 °C) and under prolonged (i.e., hours) low-energy electron irradiation[42]. The mobility of the defects observed is low, with only one edge dislocation moving over several moiré cells between measurements, after which it remained at the same position even after a month at room temperature and reheating (see Supplementary Fig. 9). This stability suggests that the moiré lattice itself could play a role in stabilizing these defects, via a minimum of the local stacking fault energy within the moiré unit cell.

These topological dislocations break translational symmetry of the moiré lattice, which may lead to singular electronic properties on the local scale[43–45]. Specifically, a phase difference

will appear between electron paths encircling the defect clockwise and counterclockwise.

**High-temperature dynamics of the moiré lattice**. All measurements presented so far were performed at 500 °C, to minimize hydrocarbon contamination under the electron beam.

In literature, there is concern about the graphene layers untwisting at such temperatures, due to energy differences between different rotations[23,24]. However, no direct observation of untwisting has been presented in the literature, apart from the relaxation observed above 800 °C in ref. [17], and theoretical work indicates untwisting of large flakes would be unlikely[46]. Indeed,

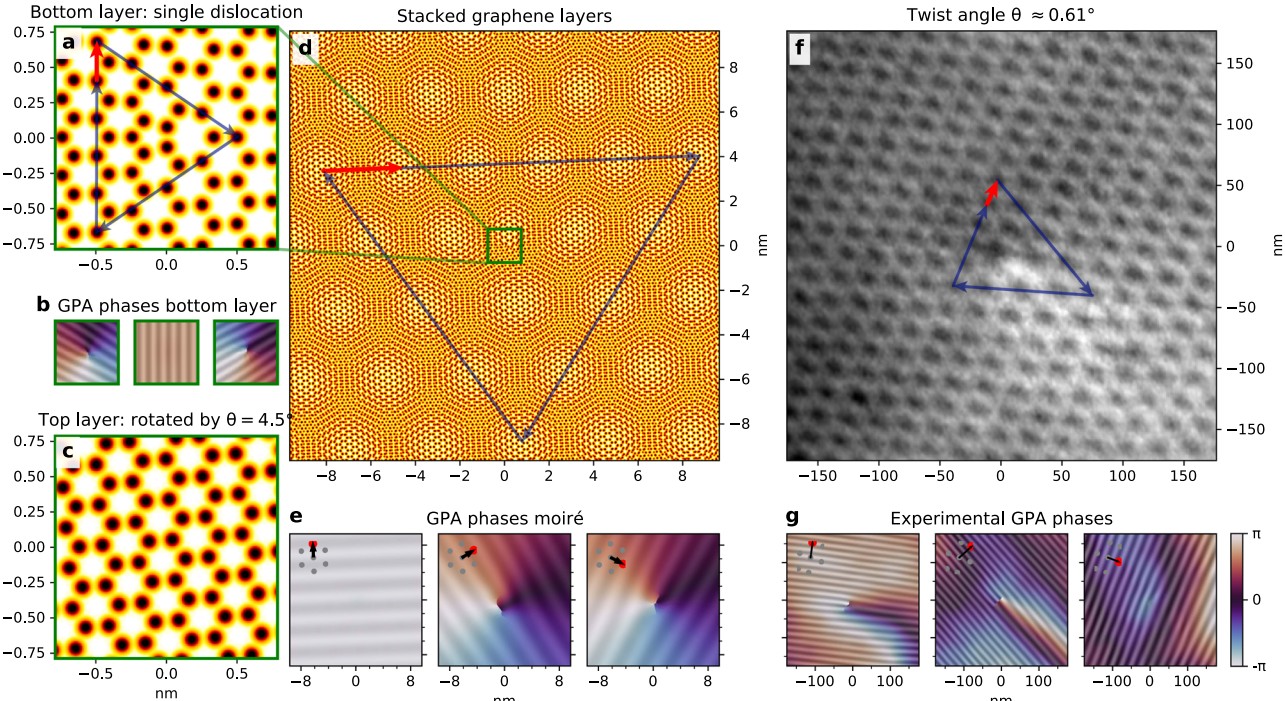

**Fig. 3 Edge dislocations in moiré systems. a** Schematic of an edge dislocation in single-layer graphene (centered in the field-of-view), with the corresponding Burgers vector analysis (blue arrows, Burgers vector indicated in red). **b** GPA phases of (**a**). **c** Top layer without dislocation, rotated 4.5° with respect to the layer in (**a**). **d** Schematic of an edge dislocation of single-layer graphene in a twisted bilayer system. The green square in the center corresponds to the combination of (**a**) and (**c**). The Burgers vector analysis of the moiré lattice defect is indicated by blue arrows, the resulting Burgers vector in red. Note: Both the schematics in (**a**) and (**d**) are mathematical representations of the dislocation, i.e., without taking atomic relaxation into account. **e** GPA phases of **d**, exhibiting a singularity in the center. The used moiré reference vectors are indicated. **f** LEEM image of an edge dislocation in a TBG moiré lattice with a twist angle $\theta \approx 0.63°$. The corresponding moiré Burgers vector is indicated in red. **g** GPA phases corresponding to (**f**).

we see no sign of untwisting. The twist angles within the domains are stable from 100 °C up to 600 °C for all samples studied, including samples with significantly larger domains between folds, such as the one studied in ref. [4]. However, in the current experiment, a full distinction between local pinning of the moiré lattice by defects and intrinsic rotational stability of large area TBG cannot be made. For that, the experiment is to be repeated on a homogeneous area, carefully isolated from the rest of the TBG sample by lithographic means.

A more subtle dynamic effect we did observe is a thermal influence on the moiré pattern. At a temperature of 500 °C, the position of the stacking domain boundaries fluctuates slightly as a function of time (see Fig. 4a–c). Taking the difference of later images (Fig. 4b, c) with the first image (Fig. 4a), we clearly see the domain boundaries shifting (Fig. 4d, e). Moreover, we can quantify these fluctuations via the difference in displacement field with respect to the image at $t = 0$ s using GPA (Fig. 4f, g). Interestingly, these involve the collective movement of millions of atoms, but only over very small distances. The full dynamics are shown in Supplementary Video 1.

We stress that a translation of the domain boundary by 4 nm, as observed, corresponds to a shift of less than half the width of the domain boundary itself[12,47]. As the relative shift of the layers over the full domain boundary is a single carbon bond length, the corresponding atomic translations are less than half of that, i.e., less than 70 pm. Hence, the "moiré magnification" makes it possible to detect these sub-angstrom changes in TBG in real-time using LEEM. Our data suggest that domain boundary displacement follows a random pattern of forward and backward steps. This indicates a possible source for the twist angle disorder observed in low(er) temperature experiments[10,13,15,33]: frozen-in

thermal fluctuations of the moiré lattice. The thermal fluctuations found, corresponding to ±0.005° for twist angle and ±0.02% for strain, are smaller than the extracted static deformations, though not negligible. Note that these values are damped by the intrinsic broadening of GPA and the time integration. Future experiments will focus on deducing the detailed statistics of the domain boundary dynamics versus temperature. Following these local collective excitations in time, will yield quantitative information on the energy landscape of these atomic lattice deformations within the moiré lattice. This will be important to answer the question whether moiré lattices can be relaxed and homogenized using controlled annealing. If so, this would yield higher-quality magic-angle TBG devices in which charge transport is not limited by percolative effects and higher critical temperatures are reached.

Our quantitative LEEM study on TBG reveals a wide variety in twist angles and strain levels in a single sample. We show that spontaneous changes in global twist angle do not occur, even at elevated temperatures, but that local collective fluctuations do take place. This suggests that high-temperature annealing causes relaxation of the local moiré lattice, reducing lattice disorder. Vice versa, this points to frozen-in thermal fluctuations as a possible source for the (significant) short-range twist angle disorder observed previously. Furthermore, this potentially offers insight into energetic aspects of the atomic lattice deformation within the moiré lattice.

We also report the observation of stable topological defects, i.e., edge dislocations, in the moiré lattice of two Van der Waals layers. Combining our methods with other techniques that can access the electronic structure, such as STS, nanoARPES, and even in situ potentiometry[48], will allow for a systematic study of the electronic properties around these defects. Finally, the

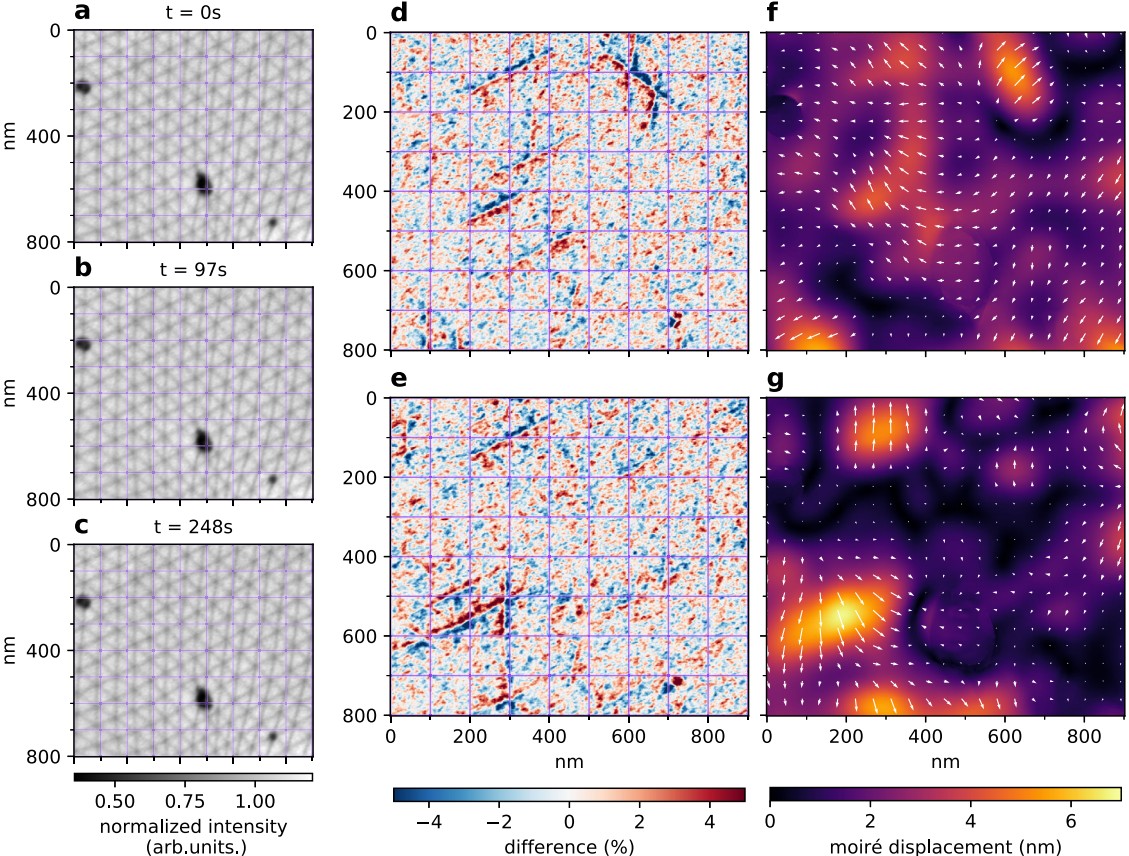

**Fig. 4 Dynamics of moiré patterns. a–c** Three images of the same area (in the same domain as Fig. 1f), taken minutes apart at a constant temperature of 500 °C. Here, $\theta^* \approx 0.18$ and $\epsilon \approx 0.04\%$ (local values as extracted by GPA shown in Supplementary Fig. 4). **d, e** Difference of respectively **b** and **c** with **a**, i.e., $t = 0$ s, highlighting the shift of the domain boundaries. **f, g** GPA extracted displacement of respectively **b** and **c** with respect to $t = 0$ s, with the arrows indicating the direction and amplitude, magnified eight times for visibility.

methods we describe here extend beyond TBG, to any type of twisted system. Therefore, our work introduces a way of studying deformations of moiré patterns and of connecting these to the (local) electronic properties of this exciting class of materials.

## Methods

**Sample fabrication**. The TBG sample was fabricated using the standard tear-and-stack method [1,49]. The monolayer graphene was first exfoliated with scotch tape onto a SiO$_2$/Si substrate. A polycarbonate (PC)/polydimethylsiloxane (PDMS) stamp was used for the transfer process, where the PC covered only half of the PDMS surface. After the first half of the graphene flake was successfully torn and picked up, it was rotated by 1.0°. The flake was then overlapped with the bottom half and used to pick it up. The stack was then stamped on a moderately thick (~140 nm) hBN flake, priorly exfoliated with PDMS on to a silicon substrate, along with the PC layer. Part of the graphene flake is deliberately put in contact with the Silicon surface for electrical contact purposes, i.e., to absorb the beam current. The whole substrate is then left in chloroform for 3 h to dissolve the PC. All flakes were exfoliated from crystals, commercially bought from HQ graphene and the fabrication process was performed using the manual 2D heterostructure transfer system sold by the same company.

**Low energy electron microscopy**. All LEEM measurements were performed in the ESCHER LEEM, based on the SPECS P90[18,19,50]. Samples were loaded into the ultrahigh-vacuum (base pressure better than $1.0 \times 10^{-9}$ mbar) LEEM main chamber and heated to 500 °C at a rate not exceeding 0.45 K per second (as measured by pyrometer and confirmed by IR-camera) and left at this temperature to get rid of any (polymer) residue (temperature log is shown in Supplementary Fig. 15). All measurements were conducted at elevated temperatures, 450–500 °C unless specified otherwise. The sample was located on the substrate using photo-emission electron microscopy with an unfiltered mercury short-arc lamp, by comparing to optical microscopy images taken beforehand. Spectra were taken in high-dynamic-range mode and drift corrected and all images were corrected for detector artifacts, as described in ref. [51]. When needed to obtain a sufficient signal-to-noise ratio, multiple 250 ms exposures were accumulated for each image, e.g., 8

exposures (2 s) per landing energy in the spectra in Fig. 1b and 16 exposures (4 s) at each location for the overview at 37 eV in Fig. 1c, e.

*Time series*. To measure the dynamics as presented in Fig. 4, a time series of accumulated $4 \times 250$ ms = 1 s exposure images were taken back-to-back. After regular detector correction and drift correction, each image was divided by a Gaussian smoothed version of itself ($\sigma = 50$ pixels) to get rid of spatial and temporal fluctuations in electron illumination intensity. To further reduce noise, a Gaussian filter with a width of $\sigma = 1$ image ~ 1 s, was applied in the time direction before applying GPA.

**Stitching**. To enable high resolution, large field-of-view LEEM imaging, the LEEM sample stage[52] was scanned in a rectangular pattern over the sample, taking an image at each position, leaving sufficient overlap (2 μm steps at a 4.7 μm field-of-view). To obtain meaningful deformation information from this, care needs to be taken to use a stitching algorithm that does not introduce additional deformation, i.e., as faithfully reproducing reality as the constituting images.

To achieve this, a custom stitching algorithm tailored towards such LEEM data was developed, as described in Supplementary Note 4 and in the implementation[53].

In addition, for the composite bright field in Fig. 1c, minor rotation and magnifications differences due to objective lens focus differences were compensated for. This was done by registering the stitches for different energies using a log-polar transformation-based method to obtain relative rotations and magnification. Subsequently, areas where a color channel was missing, were imputed using a $k$-nearest neighbor lookup in a regularly sliced subset of the area with all color channels present.

**Image analysis**. To quantify the large deviations in lattice shape due to the moiré magnification of small lattice distortions, we extended the GPA algorithm to use an adaptive grid of reference wave vectors, based on related to earlier work in laser fringe analysis[34].

The spatial lock-in signal is calculated for a grid of wave vectors around a base reference vector, converting the GPA phase to reference the base reference vector every time. For each pixel, the spatial lock-in signal with the highest amplitude is selected as the final signal. To avoid the problem of globally consistent phase

unwrapping, the gradient of each GPA phase was directly converted to the displacement gradient tensor. More details of the used algorithm are given in Supplementary Note 2.

All image analysis code was written in Python, using `Numpy`[54], `Scipy`[55], `scikit-image`[56] and `Dask`[57]. The core algorithms will be made available as an open-source Python package[35]. Throughout the development of the algorithms and writing of the paper, `matplotlib`[58,59] was extensively used for plotting and figure creation.

**Reflectivity calculations.** The theoretical reflectivity spectra are obtained with the ab initio Bloch-wave-based scattering method described in ref. [60]. Details of the application of this method to stand-alone two-dimensional films of finite thickness can be found in ref. [61]. The underlying all-electron Kohn-Sham potential was obtained with a full-potential linear augmented plane-wave method within the local density approximation, as explained in ref. [62]. Inelastic scattering is taken into account by an absorbing imaginary potential $-iV_i$, which is taken to be spatially constant ($V_i = 0.5$ eV) over a finite slab (where the electron density is non-negligible) and to be zero in the two semi-infinite vacuum half-spaces. In addition, a Gaussian broadening of 1 eV is applied to account for experimental losses.

## Data availability
The data supporting the findings of this study is available at 4TU.researchdata with the identifier DOI: 10.4121/16843510[63].

## Code availability
The analysis code is split into three parts: core algorithms are available at ref. [35], lattice rendering code used is available at ref. [64] and specific code to generate the figures in this paper is available at ref. [65].

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

## Acknowledgements

We thank Marcel Hesselberth and Douwe Scholma for their indispensable technical support. We thank D.K. Efetov and J. Aarts for getting us started on twisted bilayer graphene. We would like to thank J. Jobst and D.N.L. Kok for scientific discussions and L. Visser and G. Stam in particular for their input on the stitching algorithm. We thank Federica Galli for both scientific discussions as well as technical support. This work was supported by the Dutch Research Council (NWO) as part of the Frontiers of Nanoscience program (TAdJ, T.B.). It was also supported by the Spanish Ministry of Science, Innovation, and Universities (Project No. PID2019-105488GB-I00, E.E.K.).

## Author contributions

T.A.d.J. performed the LEEM measurements and wrote the LEEM data analysis code. X.C. fabricated the sample and performed AFM measurements. T.B. contributed to the analysis. E.E.K. performed the reflectivity calculations. S.J.v.d.M., M.J.A.d.D., R.M.T., and M.P.A. supervised the work. T.A.d.J. and S.J.v.d.M. took the lead in writing of the paper, with contributions from T.B., X.C., R.M.T., M.P.A., M.J.A.d.D., and E.E.K. All authors contributed to the scientific discussion.

## Competing interests

The authors declare no competing interests.
