## [Peer Review File · Nature Communications]

Imaging moiré deformation and dynamics in twisted bilayer grapheneREVIEWER COMMENTS

Reviewer #1 (Remarks to the Author):

The paper explores Moire lattices in twisted graphene. It uses long range measurements to focus on the Moire periodicity and understand crystallography of Moire rather than the actual individual lattice itself. It provides a new contribution to the field, which is hot at the moment. The paper is original and shows that dislocations can occur. The paper is worthy of publication after addressing a couple of minor issues.

1. Figure 1 needs some sort of introduction schematic, it jumps straight into results and the reader needs first to understand what the materials system that is being characterized. A schematic at the start of the paper helps the rest of the results to be understood easily by the general audience.

2. The discussion about dislocation movement in relation to prior TEM and ref 38 needs adjustment. Partly the authors do not cite more literature from the TEM field that gives a wrong account on dislocation stability at higher temperature. It is not correct in this current version. In TEM, it is possible to observe dislocation pairs that are spaced 1-5nm apart and form from the loss of atoms. Dislocation pairs that are close together interact by strain fields and can annihilate. but once they migrate further away from each other and the interaction is no longer present, they become stable at high temperature. The length scale of the measurements in figure 3 are big and these fine scale 1-10nm dynamics are not measured. The atomic schematic in figure 3a is also not correct, the shape of the 5-7 doesnt seem to match the correct version. The authors can use simple free software like Accllys discovery studio visualizer to make models and relax with simple force fields. Dislocations in graphene are also known to buckle the sheets. The authors should discuss this in detail and how the out of plane distortion impacts buckling and their Moire observations.

Overall, the paper has some nice results showing how dislocations in graphene impact the Moire lattice, but need to refine their understanding and description of dislocations in graphene, by including more citations in the field. There has been lots published on graphene dislocations at the atomic scale and their high temperature dynamics, but this paper doesnt accurately connect to these. Once that is tightened up, it can be published.

Reviewer #2 (Remarks to the Author):

In this paper by de Jong and co-workers, aberration-corrected Low Energy Electron Microscopy (AC-LEEM) was used to image twisted bilayer graphene (TBG) for subsequent analysis of spatial variations in twist angle and heterostrain. The authors quantified these spatial inhomogeneities in the sample and dynamics of the lattice during heating.

The authors present a combination of complementary theoretical and experimental work. Ab-initio calculations of LEEM spectra for varying stacking arrangements of TBG are used to establish an important framework for choosing the incident electron energy that gives the highest contrast between different stacking orders. The obtained LEEM spectra were then analyzed using GPA phase analysis to extract displacement fields and heterostrain maps across the sample region during heating at high temperatures of ~ 500 °C.

Discussion of the interplay between intrinsic inhomogeneity and correlated electronic phases in twisted bilayer graphene is currently of intense interest. A detailed structural analysis of twisted layers as tackled here is essential for a better understanding of exotic physics in TBG (and other

moiré materials). Thus, the topic of this paper is highly relevant for the growing field of twistronics. This paper presents structural measurements of TBG and demonstrates a powerful analysis tool that could be extended to other twisted systems. Therefore, I am confident that the work will be of interest to a broad readership in the materials sciences. I recommend this paper for publication in Nature Communications once the following comments are addressed:

1. It would be instructional if the authors commented on the applicability of this technique to samples sandwiched between two flakes of hBN, which is a common configuration for transport measurements or air-sensitive twisted 2D materials. How thin would the top-hBN need to be to allow this LEEM analysis to work?

2. The authors found that the moiré patterns are stable across a wide range of temperatures (up to 600 °C), which contrasts findings of previous literature. The question raised here is why are these samples more thermally robust than previously studied TBG samples? Is it possible that the ubiquitous folds in the sample are pinning the twisted domains? It should be possible to etch a "moat" around a flat region of the sample, creating a twisted region that is free of bubbles and folds at the periphery. Then it would be interesting to see if this more "liberated" TBG stack is also stable to heating. This would be a very valuable insight into the intrinsic thermal stability of TBG. In particular, such an experiment would set this work well apart from that of Alden et al. PNAS, 2013, 110 (28), 11256–11260 in which in situ heating and TEM was used to monitor the domain boundary fluctuations.

3. Among the primary findings of this paper is the conclusion that high temperature annealing reduces variations in local twist angle and heterostrain, but a further experiment on this would be very insightful. The authors could perform LEEM and geometric phase analysis for a freshly prepared sample at room temperature (if not possible with LEEM at room temp, then might DF-TEM be used to establish this initial state?), then upon heating to 500°C, and then again after cooling down to room temperature. This would confirm the extent to which initial inhomogeneity (as-prepared) at lower temperatures is annealed out upon heating and how much the inhomogeneities present subsequently at low temperatures arise from frozen-in thermal fluctuations of the anneal-homogenized superlattice.

4. One figure in the paper that needs additional clarification is Figure 2, where the schematic of a GPA phase analysis is presented in a) and b). In the current form, it is unclear from the figure what colors in those parts of Figure 2 represent. Thus, a color map/bar should be attached to both Figure 2a and 2b to quantify the color scale in the presented images.

5. The authors report that they can resolve the moiré pattern for twist angles up to 1.3°, but this is only shown for a 2-on-2 layer twisted graphene, while all other samples in the manuscript are twisted 0.6° or less. Can the authors comment on the possibility to achieve this level of resolution, encompassing the 'magic angle' range, for regular 1-on-1 twisted bilayer graphene?

6. Also a couple minor points: First, the rate of heating/cooling should be reported in the Methods in case this could affect the degree of homogeneity of the sample. Second, the term "annealing" is usually used with the implication that the sample has been heated and then cooled, which is not the case for the described experiments (most performed at high temperature of 500 °C).

Overall, this is an important paper that I would like to see published in Nature Communications after addressing these comments.

Reviewer #3 (Remarks to the Author):

The authors use low energy electron microscopy to image low-angle moiré patterns in twisted bilayer graphene. Theoretically calculated LEEM spectra are used to specify an electron energy which generates differential contrast from the different stacking areas present in the sample. The authors then use image stitching to map spatial variation over large areas, as well as image series to map temporal variation over small scales.

Imaging such patterns has already been demonstrated using other techniques (TEM/SEM, SPM, near-field optics). The visualisation of fluctuating domain boundaries at high temperature has not been previously reported and is an interesting result, although only one area of one sample at one temperature has been studied in this way. Using this technique to measure the dependence of the observed fluctuations on temperature and local twist angle/strain would be of great interest to those in the field.

The experimental evidence, analysis and interpretation clearly supports the conclusions of the paper. Experimental methods are clear and allow others to reproduce the reported experiment. Analytical methods (stitching, GPA, time series) are well described with the code available. The authors could expand on the benefits/limitations of LEEM imaging of moire patterns (e.g. does not require suspended samples, suitability for high T imaging) compared to previously demonstrated imaging techniques. In particular the SEM based technique should be referred to, (<https://www.nature.com/articles/s41563-020-00873-5>) which is also capable of mapping variation at large scales, even with an encapsulation layer which means it can be applied to the kind of samples where the emergent properties (superconductivity etc) mentioned have been observed.

The authors mention that there is concern in the literature about the untwisting of graphene layers at high temperatures, but the references provided (14, 15) do not seem to support that view. I would not expect large scale untwisting at elevated temperatures from these twist angles in such large samples (<https://journals.aps.org/prb/abstract/10.1103/PhysRevB.101.054109>).

Authors should specify the twist angle for the sample shown in figure 4. It would be instructive to present a map of heterostrain and local twist for the area shown in this figure/ supplementary video 1. Can the authors quantify the level of fluctuations across the sample (spatially and temporally) during the video? Are certain areas more or less stable to deformation?

It would be helpful if the speed of supplementary video 1 could be reduced – it is quite hard to track the changes at the current speed. Figures would be clearer with spatial axes labels removed.

Dear reviewers,

We thank the reviewers for their detailed and constructive comments and are convinced their expertise input, in particular on prior TEM results and broader context, helped create a better manuscript.

A point-by-point response to comments is included below.

With kind regards on behalf of all authors,

Tobias A. de Jong and Sense Jan van der Molen

Reviewer #1 (Remarks to the Author):

The paper explores Moire lattices in twisted graphene. It uses long range measurements to focus on the Moire periodicity and understand crystallography of Moire rather than the actual individual lattice itself. It provides a new contribution to the field, which is hot at the moment. The paper is original and shows that dislocations can occur. The paper is worthy of publication after addressing a couple of minor issues.

We thank the reviewer for their positive comments.

1. Figure 1 needs some sort of introduction schematic, it jumps straight into results and the reader needs first to understand what the materials system that is being characterized. A schematic at the start of the paper helps the rest of the results to be understood easily by the general audience.

We agree with the reviewer and added a schematic introducing the device layout and the general concept of a moiré pattern (new Fig 1a).

2. The discussion about dislocation movement in relation to prior TEM and ref 38 needs adjustment. Partly the authors do not cite more literature from the TEM field that gives a wrong account on dislocation stability at higher temperature. It is not correct in this current version. In TEM, it is possible to observe dislocation pairs that are spaced 1-5nm apart and form from the loss of atoms. Dislocation pairs that are close together interact by strain fields and can annihilate. but once they migrate further away from each other and the interaction is no longer present, they become stable at high temperature. The length scale of the measurements in figure 3 are big and these fine scale 1-10nm dynamics are not measured.

The text has been updated to better reflect the existing literature on TEM work on dynamics of edge dislocations in single layer graphene. As a result of the much lower energies in LEEM compared to TEM, electron sputtering and bond rotations needed for edge dislocation creation and movement are much less likely to be induced by the LEEM beam. However, the main text now acknowledges that no conclusions on dislocation dynamics at length scales below the moiré length scale can be drawn from the present work

The atomic schematic in figure 3a is also not correct, the shape of the 5-7 doesnt seem to match the correct version. The authors can use simple free software like Acclrys discovery studio visualizer to make models and relax with simple force fields.

Figure 3a-d are simple schematics meant to explain the experimental observation. Hence, we decided to show a representation of a hexagonal lattice with a mathematical dislocation instead of a fully relaxed physical model. We do understand the criticism of the reviewer on the appearance of the original version however, and have updated the figure and added a note that the schematic is a representation of a mathematical dislocation.

Dislocations in graphene are also known to buckle the sheets. The authors should discuss this in detail and how the out of plane distortion impacts buckling and their Moire observations.

Dislocations in freestanding graphene are indeed known to buckle the sheet. In this case, the second layer and substrate squeeze out most 3D deformation, as indicated by the AFM measurements in Supplementary Figure 11, which show no sign of buckling (within the error of that measurement).

Overall, the paper has some nice results showing how dislocations in graphene impact the Moire lattice, but need to refine their understanding and description of dislocations in graphene, by including more citations in the field.

There has been lots published on graphene dislocations at the atomic scale and their high temperature dynamics, but this paper doesnt accurately connect to these. Once that is tightened up, it can be published.

We thank the referee again and have indeed added references to previous studies on dislocations in single layer graphene. We do however reiterate that as far as we know this is the first work on dislocations in twisted van der Waals systems such as the twisted bilayer graphene discussed here.

Reviewer #2 (Remarks to the Author):

In this paper by de Jong and co-workers, aberration-corrected Low Energy Electron Microscopy (AC-LEEM) was used to image twisted bilayer graphene (TBG) for subsequent analysis of spatial variations in twist angle and heterostrain. The authors quantified these spatial inhomogeneities in the sample and dynamics of the lattice during heating.

The authors present a combination of complementary theoretical and experimental work. Ab-initio calculations of LEEM spectra for varying stacking arrangements of TBG are used to establish an important framework for choosing the incident electron energy that gives the highest contrast between different stacking orders.

The obtained LEEM spectra were then analyzed using GPA phase analysis to extract displacement fields and heterostrain maps across the sample region during heating at high temperatures of ~500 °C.

Discussion of the interplay between intrinsic inhomogeneity and correlated electronic phases in twisted bilayer graphene is currently of intense interest. A detailed structural analysis of twisted layers as tackled here is essential for a better understanding of exotic physics in TBG (and other moiré materials). Thus, the topic of this paper is highly relevant for the growing field of twistrionics. This paper presents structural measurements of TBG and demonstrates a powerful analysis tool that could be extended to other twisted systems. Therefore, I am confident that the work will be of interest to a broad readership in the materials sciences. I recommend this paper for publication in Nature Communications once the following comments are addressed:

We thank the reviewer for their positive comments.

1. It would be instructional if the authors commented on the applicability of this technique to samples sandwiched between two flakes of hBN, which is a common configuration for transport measurements or air-sensitive twisted 2D materials. How thin would the top-hBN need to be to allow this LEEM analysis to work?

Indeed, we do believe that observation of the moiré pattern through a top layer of hBN of at most a few atomic layers thick should be possible using this LEEM technique. In Supplementary Figure 3, a moiré pattern is observed on a 2-on-2 twisted graphene system, already showing that moiré patterns can be clearly observed through an additional atomic layer. Furthermore, we have data where a moiré pattern between a bottom hBN and a graphene layer is visible through 4 additional layers of graphene (attached). Although both these systems are slightly different from TBG covered with hBN, they do demonstrate the principal possibility of observing moiré patterns in more deeply lying layers. A paper in which we more thoroughly explore moiré contrast through different layers is subject of a future publication in preparation.

A sentence about the potential to apply LEEM measurements to devices with a thin top hBN layer was added at the end of the introduction.

2. The authors found that the moiré patterns are stable across a wide range of temperatures (up to 600 oC), which contrasts findings of previous literature. The question raised here is why are these samples more thermally robust than previously studied TBG samples? Is it possible that the ubiquitous folds in the sample are pinning the twisted domains?

Although the ubiquitous folds certainly play a role in pinning the domains, as already indicated by the strain increasing near the folds (as described in the main text), we expect that the stability extends to more general samples, including samples with much lower fold density.

As clarified in the main text, thermal stability was observed for multiple samples from different sources (Leiden University as well as ICFO, Dmitri K. Efetov lab). We are not aware of any direct observation of untwisting at these temperatures reported in literature, suggesting these samples are probably not much more thermally robust than others.

It should be possible to etch a “moat” around a flat region of the sample, creating a twisted region that is free of bubbles and folds at the periphery. Then it would be interesting to see if this more “liberated” TBG stack is also stable to heating. This would be a very valuable insight into the intrinsic thermal stability of TBG. In particular, such an experiment would set this work well apart from that of Alden et al. PNAS, 2013, 110 (28), 11256—11260 in which in situ heating and TEM

was used to monitor the domain boundary fluctuations.

Alden et al. report domain boundary motion “below 800C is rare”, corresponding to the temperature range explored in this work. Note that they don’t measure domain boundary fluctuations as done in the current work, but only relaxation at an integration time of 5*20 seconds per frame.

We do appreciate the suggestion of the referee for the method to more systematically study relaxation in a sample better that better approaches ‘intrinsic’ conditions. We will consider such experiments, but these are time-consuming and well beyond the scope of the current paper.

A reference to Alden et al. was added to the manuscript.

3. Among the primary findings of this paper is the conclusion that high temperature annealing reduces variations in local twist angle and heterostrain, but a further experiment on this would be very insightful.

The authors could perform LEEM and geometric phase analysis for a freshly prepared sample at room temperature (if not possible with LEEM at room temp, then might DF-TEM be used to establish this initial state?), then upon heating to 500°C, and then again after cooling down to room temperature. This would confirm the extent to which initial inhomogeneity (as-prepared) at lower temperatures is annealed out upon heating and how much the inhomogeneities present subsequently at low temperatures arise from frozen-in thermal fluctuations of the anneal-homogenized superlattice.

We thank the reviewer for this suggestion for a future experiment. Doing this in LEEM at room temperature is complicated, however, as it is very hard to avoid buildup of contamination at the landing energies needed to image the moiré, preventing further measurements at higher temperatures. This would hold even more strongly for DF-TEM measurements, which would also require a different substrate.

4. One figure in the paper that needs additional clarification is Figure 2, where the schematic of a GPA phase analysis is presented in a) and b). In the current form, it is unclear from the figure what colors in those parts of Figure 2 represent. Thus, a color map/bar should be attached to both Figure 2a and 2b to quantify the color scale in the presented images.

We thank the referee for bring this omission to our attention. A color bar indicating the quantitative meaning of the colors in both 2a and 2b is added.

5. The authors report that they can resolve the moiré pattern for twist angles up to 1.3°, but this is only shown for a 2-on-2 layer twisted graphene, while all other samples in the manuscript are twisted 0.6° or less. Can the authors comment on the possibility to achieve this level of resolution, encompassing the ‘magic angle’ range, for regular 1-on-1 twisted bilayer graphene?

The stated twist angle of up to 1.3 degree was also achieved on regular 1-on-1 TBG, more precisely on the sample studied in <https://doi.org/10.1038/s41567-020-01041-x>. However, this was only realized after observing the moiré on later samples and carefully reinspecting the older data. As such, the data taken on that sample has incorrect stigmatism and focus, making it unfortunately unpublishable.

6. Also a couple minor points: First, the rate of heating/cooling should be reported in the Methods in case this could affect the degree of homogeneity of the sample.

The heating rate has been added to the Methods and a graph of the temperature during heating has been added to the supplementary information as Supplementary Figure 15.

Second, the term “annealing” is usually used with the implication that the sample has been heated and then cooled, which is not the case for the described experiments (most performed at high temperature of 500 °C).

The referee is correct, and we replaced “annealing” with heating when describing our experiments to avoid confusion. We do however feel our results have implications for annealing and subsequently measuring at cryogenic temperatures with other probes as well, so it is left in the manuscript where appropriate.

Overall, this is an important paper that I would like to see published in Nature Communications after addressing these comments.

Reviewer #3 (Remarks to the Author):

The authors use low energy electron microscopy to image low-angle moiré patterns in twisted bilayer graphene. Theoretically calculated LEEM spectra are used to specify an electron energy which generates differential contrast from the different stacking areas present in the sample. The authors then use image stitching to map spatial variation over large areas, as well as image series to map temporal variation over small scales.

Imaging such patterns has already been demonstrated using other techniques (TEM/SEM, SPM, near-field optics). The visualisation of fluctuating domain boundaries at high temperature has not been previously reported and is an interesting result, although only one area of one sample at one temperature has been studied in this way.

We have observed fluctuating domain boundaries regularly, on different areas. We added several results to the Supplementary information as Supplementary Figure 6,7.

Using this technique to measure the dependence of the observed fluctuations on temperature and local twist angle/strain would be of great interest to those in the field.

The experimental evidence, analysis and interpretation clearly supports the conclusions of the paper. Experimental methods are clear and allow others to reproduce the reported experiment. Analytical methods (stitching, GPA, time series) are well described with the code available.

We thank the reviewer for their positive comments.

The authors could expand on the benefits/limitations of LEEM imaging of moire patterns (e.g. does not require suspended samples, suitability for high T imaging) compared to previously demonstrated imaging techniques.

In particular the SEM based technique should be referred to, (<https://www.nature.com/articles/s41563-020-00873-5>) which is also capable of mapping variation at large scales, even with an encapsulation layer which means it can be applied to the kind of samples where the emergent properties (superconductivity etc) mentioned have been observed.

We were aware of the SEM based technique and apologize for the omission. The reference was added, in combination with a more extensive discussion of upsides and downsides of LEEM for the imaging of moiré patterns at the end of the Introduction.

The authors mention that there is concern in the literature about the untwisting of graphene layers at

high temperatures, but the references provided (14, 15) do not seem to support that view. I would not expect large scale untwisting at elevated temperatures from these twist angles in such large samples (<https://journals.aps.org/prb/abstract/10.1103/PhysRevB.101.054109>).

We were not aware of the theoretical work referred to by the reviewer and are glad to include a reference to it. Although we are indeed not aware of any direct observations of untwisting in literature, we did encounter severe concerns in the community that untwisting could happen, both from discussions and in literature. Although the main text of Ref. 14 does not report untwisting, the SI says: "During device fabrication special care is taken so that the temperature of the sample never exceeds 150 °C to avoid untwisting of the TBG".

Most people we discussed this rather practical matter with, referred to the sample preparation procedure in ref 15, which explicitly mentions low annealing temperatures.

The first paragraph on dynamics was updated to clarify and to include the mentioned reference.

Authors should specify the twist angle for the sample shown in figure 4. It would be instructive to present a map of heterostrain and local twist for the area shown in this figure/ supplementary video 1.

The area shown in Figure 4 is in the same domain as Fig 1e and included in the purple histogram in Fig 2f,g.

Text was added in the caption to clarify this and a separate Supplementary Figure 4 showing both local twist angle and heterostrain of the area shown in the supplementary movie was added.

Can the authors quantify the level of fluctuations across the sample (spatially and temporally) during the video?

The mean absolute displacement during the video fluctuates between 0.8 and 1.5 nm, with an average of about 1 nm. Spatially, some areas show significantly more variation, with some areas reaching an average displacement from the mean position of more than 2.5nm over the duration of the video. A figure with the mean absolute displacement relative to the average deformation as a function of time and the average displacement relative to the mean as a function of position has been added to the Supplementary information as Supplementary Figure 5.

Are certain areas more or less stable to deformation?

This is a question of great interest. From what we have observed, it seems domain boundaries spaced farther apart (i.e. in lower twist angle areas) are on average more mobile, but this is confounded by the moiré magnification increasing for lower twist angles. A more quantitative investigation of a larger dataset is needed to draw firm conclusions on this. We do intend to pursue this in the future.

It would be helpful if the speed of supplementary video 1 could be reduced – it is quite hard to track the changes at the current speed.

We have slowed down the video. Additionally, Figure 4 and the movie have been updated to better include the direct calculation of the gradient (as described in the SI).

Figures would be clearer with spatial axes labels removed.

We prefer this method over the use of scalebars since we believe it is more clear and it does not obstruct the presented images. However, if the editors prefer scalebars instead we are willing to change to scalebars.

REVIEWERS' COMMENTS

Reviewer #1 (Remarks to the Author):

I was positive on this paper in the first round of review. I have no further requested changes or comments. I am happy to support publication in its current form.

Reviewer #2 (Remarks to the Author):

The authors have modified the article to address some of the issues raised.

One key issue that I think still warrants an important caveat in the text is the point that this stability of the moiré pattern to thermal treatment is observed in samples here with folds and bubbles that can pin the layers globally (even if the bubbles are far apart). I appreciate the authors' point that making a more isolated TBG sample with the "moat" structure that I suggested would be technically challenging. However, I think that is the way to unambiguously experimentally evaluate whether or not the twist is intrinsically stable to heating. Since this experiment may be too challenging to accomplish in the context of this particular report, I think a statement on this point is needed at the very least, to articulate this caveat and prompt future experiments on this.

After such an important discussion (short) is incorporated, I can recommend the paper for publication in Nature Communications.

Reviewer #3 (Remarks to the Author):

The changes in response to reviewer feedback are reasonable and I believe the paper is now suitable for publication.

We thank all reviewers for their comments. Regarding the remaining point of Reviewer #2, we have appended the following to the observation of the lack of untwisting in the main text:

“However, in the current experiment a full distinction between local pinning of the moiré lattice by defects and intrinsic rotational stability of large area twisted bilayer graphene cannot be made. For that, the experiment is to be repeated on a homogeneous area, carefully isolated from the rest of the TBG sample by lithographic means.”